# Heteroleptic Complexes of Ruthenium Nitrosyl with Pyridine and Bypiridine—Synthesis and Photoisomerization

**DOI:** 10.3390/molecules29174039

**Published:** 2024-08-26

**Authors:** Anastasiya O. Brovko, Natalya V. Kuratieva, Denis P. Pishchur, Gennadiy A. Kostin

**Affiliations:** Nikolaev Institute of Inorganic Chemistry Siberian Branch of Russian Academy of Science, Lavrentieva 3, 630090 Novosibirsk, Russia; a.brovko@g.nsu.ru (A.O.B.); kuratieva@niic.nsc.ru (N.V.K.); denispishchur@yandex.ru (D.P.P.)

**Keywords:** ruthenium complex, nitrosyl complex, photoisomerization, bipyridine

## Abstract

The reaction of [RuNO(Py)_2_Cl_2_OH] with bipyridine in water–ethanol media results in trans-(NO, OH)-[RuNO(Py)(Bpy)ClOH]^+^ with an acceptable yield (60–70%) as hexafluorophosphate salt. Further treatment of the hydroxy-complex with concentrated HF quantitatively leads to trans-(NO, F)-[RuNO(Py)(Bpy)ClF]^+^. Despite the chirality of both coordination spheres, the hexafluorophosphate salts crystallized as racemates. A NO-linkage isomerism study of the obtained complexes was performed at 80 K with different excitation wavelengths (405, 450, 488 nm). The most favorable wavelengths for the MS1 isomer (Ru-ON) formation were 405 and 450 nm, where the linkage isomer populations were 17% and 1% for [RuNO(Py)(Bpy)ClOH]PF_6_ and [RuNO(Py)(Bpy)ClF]PF_6_. The shift of the excitation wavelength to the green (488 nm) sharply decreased the MS1 population. The IR-spectral signatures of MS1 were registered. Reverse-transformation Ru-ON (MS1)-Ru-NO (GS) was investigated for [RuNO(Py)(Bpy)ClOH]PF_6_ using IR and DSC techniques that made it possible to determine the kinetic parameters (E_a_ and k_0_) and decay temperature.

## 1. Introduction

Ruthenium complexes with polydentate heterocycles (bipyridine, phenanthroline, terpyridine, etc.) are widely explored due to their luminescence and electrochemical properties [1,2,3]. The CCDC database contains almost 4000 structures of ruthenium complexes with polypyridines; most of them are ruthenium (II) complexes. Far fewer examples are known for ruthenium nitrosyl complexes (Ru-NO) with that type of ligand. The first publication about ruthenium nitrosyls with polypyridines was in 1971 by T.J. Meyer and J.B. Godwin. They reported the synthesis of [RuNOL_2_NO_2_]^2+^ (L = bpy, phen) from RuNOL_2_Cl_2_ complexes [4]. The first structurally characterized species were prepared at the end of the 20th century in the Mukaida group [5,6]. The growing interest in those complexes at the beginning of the 21st century was determined by the fact that the Ru-NO group can undergo photoisomerization in the solid state and photo-dissociation in a solution [7,8,9,10]. In both cases, the driving force of the process is the photoactivation of the Ru-NO bond through the charge transfer from the HOMO (d-orbitals of ruthenium or upper-lying π-bonding or p-non-bonding orbitals of ligands) to the LUMO with the π* (Ru-NO) contribution. The weakening of the Ru-NO bond results in free NO release in the solution, while in the solid state, the rotation of the ligand results in the formation of the Ru-ON (MS1) and Ru-η_2_ (NO) (MS2) metastable isomers. Polypyridine ligands are good antennas for irradiation due to the large π-system and can decrease the HOMO–LUMO gap; thus, the excitation wavelengths can be shifted from the blue to the red region. This is why nitrosyl complexes with similar ligands were mostly investigated in solutions to increase the quantum yield of Ru-NO with excitation in the therapeutic window (500–1000 nm). A large variety of terpyridine derivatives of ruthenium nitrosyl cis- and trans-(Cl,Cl)-[RuNO(R-Tpy)Cl_2_]^+^ (R-Tpy = 4′-R-2,2′:6′,2″-terpyridine) were prepared in the I. Malfant group [11,12,13,14]. The NO-release quantum yields were larger for the cis complexes, but correlations with the donating ability of organic ligands were not found. Additionally, two-photon absorption accompanied by NO release was detected upon excitation with 810 nm light. The systematic investigation of ruthenium nitrosyls carrying polypyridines with carboxy-amide moieties was performed in the P. Mascharak group [15,16,17,18]. The most efficient dye–nitrosyl conjugates exhibited NO release, even with 600 nm photoexcitation. 

The photoisomerization of ruthenium nitrosyl complexes with olygopyridines has been studied less extensively. A comparison of the different pyridine complexes trans-(NO,X) [RuNO(Py)_4_X]^2+^ with bipyridine analogues trans-(NO,X)-[RuNO(Bpy)_2_X]^n+^ (X = Cl, n = 2, X = H_2_O, n = 3) and trans-(NO,N) [RuNO(Tpy)Cl_2_]^+^ revealed the decrease in the achieved population of MS1 state that is the most pronounced for the terpyridine complex [19]. The isomerization of cis-(NO,OH)-[RuNO(Tpy)ClOH]PF_6_ was examined using the DSC technique after photoexcitation with a metal vapor lamp with a 410–500 nm filter [20]. The resulting populations of the MS1 and MS2 states were very low compared to Na_2_[FeNO(CN)_5_]; the MS1 and MS2 decay temperatures were 180 and 147 K. An investigation of [RuNO(Tpy)(Glycolate)]BPh_4_ using 488 nm photoexcitation revealed a 9.5% MS1 population, and the isomerization was accompanied by NO release from the sample [21]. Surprising results were obtained for cis-(NO,NO_2_)-[RuNO(Bpy)_2_NO_2_](PF_6_)_2_; the photoisomerization of the Ru-NO group is accompanied by intramolecular oxygen transfer from NO_2_ to the NO ligand, resulting in an isonitrosyl-nitrito isomer [22].

Previously, we investigated the influence of the fluoride ligand on the thermal stability of ruthenium nitrosyl linkage isomers [23,24,25]. In the present manuscript, we describe two new complexes [RuNO(Py)(Bpy)ClX]PF_6_ (X = OH, F) and investigated their photoisomerization in the solid state. 

## 2. Results and Discussion

### 2.1. Synthesis and Structure

[RuNO(Py)_2_Cl_2_OH] was chosen for the synthesis of [RuNO(Py)(Bpy)ClX]PF_6_ (X = OH, F) heteroleptic complexes with trans NO-Ru-X coordinates because the reaction of RuNOCl_5_^2-^ with bidentate amines results in a mixture of isomers, including one with the trans-NO-Ru-N coordinate. In the range of the molar ratio bpy:Ru = 1.5–2, the main product formed in the water–ethanol mixture was the target hydroxy complex [RuNO(Py)(Bpy)ClOH]^2+^, with the formation of [RuNO(Bpy)_2_OH]^2+^ not being observed. The increase in the bipyridine molar excess led to the increased formation of bright-red [Ru(Bpy)_3_]^2+^, which was quite unexpected for the chemistry of stable complexes of ruthenium nitrosyl. The corresponding salt [Ru(Bpy)_3_](PF_6_)_2_ can be removed from the target complex, [RuNO(Py)(Bpy)ClOH]PF_6_, by washing with methylene chloride. This is why the reaction conditions for the target complex preparation were restricted by a Bpy:Ru ratio in the range of 1.5–2. The subsequent replacement of the OH group with the fluoride ligand in the trans position to the NO occurred easily enough in concentrated HF. The completeness of the reaction could be monitored using the IR spectra of products. The band of the nitrosyl group for the hydroxy complex [RuNO(Py)(Bpy)ClOH]^2+^ (1852 cm^−1^) disappeared completely, and a new band appeared, with a significant shift to the high-energy region (1917 cm^−1^), corresponding to the trans coordinate NO-Ru-F. 

Both cations of [RuNO(Py)(Bpy)ClX]^2+^ (X = OH, F) are chiral, with an asymmetric center in the ruthenium atom. The salt [RuNO(Py)(Bpy)ClOH]PF_6_ crystallized in the centrosymmetric group P2_1_/n as the racemate. The salt [RuNO(Py)(Bpy)ClF]PF_6_ crystallized in the non-centrosymmetric group P2_1_2_1_2_1_; however, the BASF parameter during structure refinement is very close to 0.5 (0.489), which indicates the formation of racemate. 

The crystal structures of the complexes are formed by the ruthenium octahedral [RuNO(Py)(Bpy)ClX]^2+^ cations (Figure 1) with hexafluorophosphate anion. The heteroaromatic and chlorine ligands of the ruthenium cations are located in the cis-position to the nitrosyl ligand. The trans-coordinates of the obtained complexes differ with either hydroxyl or fluorine in the trans-position to nitrosyl ligand. The N-O and Ru-NO bond lengths are typical for ruthenium nitrosyl complexes, displaying an almost linear Ru(II)-NO^+^ core [26]. The main bond distances and angles for both complexes can be found in Table 1.

#### 2.1.1. [RuNO(Py)(Bpy)ClOH]PF_6_

The ruthenium octahedron is slightly distorted, with the angle O(OH)-Ru-N(NO) being close to 180° (176.45°). The angles N-Ru-OH and Cl-Ru-OH between equatorial ligands and hydroxyl group are close to 90° (85.54° and 85.17° for bipyridine, 88.12° for pyridine and 88.51° for chlorine). The pyridine ring is rotated to the equatorial RuN_3_Cl plane at an angle of 56.07° due to the spatial constraints. 

The Hirshfeld surface analysis shows that in both structures, the intermolecular interactions of cations primarily involve the non-specific interaction with hydrogen atoms (Appendix A). The stacking interactions between aromatic rings and interactions of the NO group with PF_6_^−^ counterions are also significant.

In RuNO(Py)(Bpy)ClOH]PF_6_, the couple of two different chiral isomers form an interesting structural element (Figure 2a). The bipyridine ligands of adjacent cations exhibit parallel-offset interactions, with the shortest distance between carbon atoms being 3.455 Å (the distance between bipyridine centroids is 3.745 Å), demonstrating a parallel-offset type of aromatic stacking [27,28]. The aromatic rings of the pyridine ligands are positioned on the top and at the bottom of the two parallel bipyridines. The planes of the pyridine rings deviate from the bipyridine planes by an angle of 9.53°, the shortest intermolecular distances between carbon atoms of pyridine and bipyridine ligands being equal to 3.516 Å. These arrangement creates the unified offset-stacking system for all four cations (Figure 2a).

Hexafluorophosphate anions are primarily located in the hydrophilic part of the crystal structure (Figure 2b) combining the cations in the chains along (1 0 −1) direction. The distances between the fluorine atoms and the oxygen (nitrogen) atoms of nitrosyl groups are shorter than the sum of the Van der Waals (VdW) radii (2.99–3.02 Å). The distance between the fluorine and the oxygen atom of hydroxyl group is 3.119 Å, indicating a weak hydrogen-bonding interaction. 

#### 2.1.2. [RuNO(Py)(Bpy)ClF]PF_6_

The ruthenium octahedron is slightly distorted, with the angle F-Ru-N(NO) being close to 180° (178.61°). The angles of equatorial ligands to the fluorine group are close to 90° (84.22° and 85.44° for bipyridine, 87.03° for pyridine, and 88.97° for chlorine). The planes of organic aromatic ligands are at an angle of 53.26° due to spatial constraints. 

Short intermolecular contacts in the structure of [RuNO(Py)(Bpy)ClF]PF_6_ are primarily formed by the hexafluorophosphate anions (Figure 3a). The main contacts are with the nitrosyl group (2.812 Å and 2.894 Å to oxygen and 2.921 Å and 2.988 Å to nitrogen from different fluorine atoms) and with the face of a bipyridyl ligand (3.141 Å and 3.155 Å to bridging carbon atoms). The interactions between PF_6_^−^ and bipyridyl ligands prevent the formation of stacking interactions (Figure 3b). Although the shortest distance between bipyridyl planes is 3.192 Å, the parallel offset between centroids (7.761 Å) is too distant to provide π stacking.

Thus, despite the similarity of cation units (Table 1, Figure 1), the overall crystal structure and intermolecular interactions in **1** and **2** differs significantly. In both structures, the PF_6_^−^ anions are located near the nitrosyl group of ruthenium cation with F…N and F…O contacts shorter than the sum of VdW radii. Weak interactions between PF_6_^−^ and bipyridyl ligands in [RuNO(Py)(Bpy)ClF]PF_6_ prohibit the formation of π stacking, while in [RuNO(Py)(Bpy)ClOH]PF_6_, the anion participates in weak hydrogen bonding with the OH-group of the ruthenium cation.

### 2.2. Photogeneration of Linkage Isomers in Solid State

Due to their relatively low thermal stability, the metastable linkage isomers were photogenerated at 80 K and characterized using IR-spectroscopy. The ν(NO) band is extremely sensitive to changes in the NO-linkage type, thus providing information on the type and population of the different linkage isomers (GS, MS1, or MS2). In GS, at 80 K, both complexes exhibit the NO-vibration band at the same positions as at room temperature. Upon irradiation of the samples, new ν(NO) bands appear: 1714 cm^−1^ for [RuNO(Py)(Bpy)ClOH]PF_6_ and 1768 cm^−1^ for [RuNO(Py)(Bpy)ClF]PF_6_ (Figure 4). The change of the linkage from Ru-NO (GS) to Ru-ON (MS1) results in a typical downshift of the ν(NO) band by 130–150 cm^−1^ [29]. The generation of the MS2 isomer is usually performed by IR irradiation of MS1 since the UV spectra of GS and MS2 strongly overlap and the MS2-MS1 photochemical equilibrium is shifted to the MS1 isomer in the blue region of irradiation. However, some examples of MS2 formation together with MS1 after blue-light irradiation of GS were detected [30]. Due to structural features, the shift of the ν(NO) band in the GS-MS2 transition (270–350 cm^−1^) is larger than in the GS-MS1 transition (130–150 cm^−1^), and no corresponding changes in IR-spectra were found in the investigated complexes.

Irradiation by different LEDs (405, 450, and 488 nm) with equal optical power reveals that the irradiation with 405–450 nm leads to approximately similar conversion toward MS1 isomers in **1** and **2**, reaching a maximum population of 17% and 2%, respectively. The populations of the metastable states were estimated based on the area under the ν(NO)_GS_ band before and after light irradiation. Further shifting to the green region (488 nm) results in a drastic decrease in the MS1 population for complex **1** and a complete disappearance of the MS1 band for complex **2**. Since the achieved population is determined by the photochemical equilibrium ruled by the absorption coefficients for corresponding linkage isomers, such behavior was expected for pyridine (bipyridine) complexes. Indeed, the known examples of MS1 isomers for pyridine complexes have absorption, namely, in the green region [31]. Since the change of the NO coordination mode leads to changes in bond lengths for other ligands [23,32], the changes in IR-spectra were also detectable in the ν(OH) vibration modes. An additional ν(OH) band corresponding to the MS1 isomer in the {RuNO(Py)(Bpy)ClOH]PF_6_ arises at 3533 cm^−1^, shifted from 3556 cm^−1^ in the ground state. Further heating of the irradiated samples results in the disappearance of ν(NO)_MS_. For complex **1**, where the population of MS1 is relatively large, the isothermal kinetic of reverse MS1-GS transformation was investigated by IR. The irradiated sample was heated to the necessary temperature, and the decay of ν(NO)_MS_ was registered (Figure 5). The kinetic curves were treated by the first law decay equation I = I_max_exp (−kt) + I_0_, and corresponding values of efficient rate constants were linearized in Arrhenius coordinates, resulting in activation parameters E_a_ = 66.0 ± 1.4 kJ/mol, lgk_0_ = 14.4 ± 0.36. 

Since the population of MS1 in [RuNO(Py)(Bpy)ClF]PF_6_ was low, the reliable heat effects during the reverse transition of MS1-GS were achieved only for the [RuNO(Py)(Bpy)ClOH]PF_6_. DSC curves registered with different heating rates are given on Figure 6, along with calculated curves according to the first-order kinetic law. Although the maximum of exothermic effects (T_max_) regularly shifts to higher temperatures with an increase in the heating rate (b), all three curves were satisfactorily described by one set of activation parameters E_a_ = 67.5 ± 4 kJ/mol, lgk_0_ = 14.4 ± 2. Within the limits of uncertainty, the same results can be obtained from the Kissenger approximation [33,34] of the dependences between T_max_ and b. The Kissenger method is a model-free approximation derived from the main kinetic law:$$W=\frac{d\alpha }{dt}=k_{0}\exp\left(-\frac{E_{a}}{RT}\right)f(\alpha )$$
where α is the extent of the conversion of the reagent to the product. Taking into account the linear dependence between t (time) and T (temperature) in DSC, the equation can be transformed to
$$\ln\left(\frac{T_{max}^{2}}{b}\right)=\frac{E_{a}}{RT_{max}}+\ln(\frac{E}{k_{0}\cdot R})$$

This allows for the independent determination of E_a_ = 66.5 ± 2 kJ/mol, lgk_0_ = 14.2 ± 0.8. 

Both the data from the IR kinetic investigation and the DSC investigation show similar results for the activation parameters of the MS1-GS transformation in the [RuNO(Py)(Bpy)ClOH]PF_6_ complex. The decay temperature previously determined as the temperature at which the decay constant is equal 10^−3^ s^−1^ [35] for [RuNO(Py)(Bpy)ClOH]PF_6_ calculated from activation parameters is equal to 198 K (IR) and 202 K (DSC), which is close to the onset of exothermal effects on the DSC curves. 

### 2.3. Molecular Orbitals and Electronic Transitions 

To understand the reason for the significant difference in the achieved population for the MS1 linkage isomer between hydroxy- and fluoro-complexes, DFT calculations were performed. For both compounds, the calculated frequences of ν(NO) correspond well to the experimental ones (1920 and 1917 for **2**, 1857 and 1851 for **1**), indicating the accuracy of the calculation for these complexes. The general appearance of the electronic spectra weakly depends on the nature of the trans-to-NO ligand (OH vs. F), and the calculated electronic spectra resolve the main features of the experimental ones well (Figure 7). 

The experimental absorption coefficients of [RuNO(Py)(Bpy)ClOH]PF_6_ on the UV-edge (300–450 nm) are slightly higher than those of [RuNO(Py)(Bpy)ClF]PF_6_. The most intense transitions in the spectra of both compounds lie in the UV region (190–300 nm) and mostly correspond to π-π* excitations in the arene rings. Much weaker broad shoulders observed in the range 350–500 nm correspond to the transitions from HOMO-HOMO-4 to LUMO-LUMO + 3 (Table 2). 

The necessary conditions for irradiation to efficiently convert GS to metastable linkage isomers were summarized earlier [36]. Firstly, the excitation wavelength should correspond to the charge transfer from metal d-orbitals participating in HOMO to LUMO with antibonding π* (-Ru-NO) character (metal-to-ligand charge transfer, MLCT). This results in the weakening of the Ru-NO bond in the exited state, allowing the rotation of NO. Second, the MLST band for the ground state should stay far enough from the MLCT band in the linkage isomers to prohibit the reverse photo-induced transformation with the same wavelength. Experimental data for different complexes [23,24,37,38] show that the second requirement is fulfilled well for GS-MS1 transformations due to the large difference in the electronic spectra of these states. Thus, the difference can be determined by the nature of occupied and virtual orbitals participating in the ground state photoexcitation. 

The nature of LUMO-LUMO + 3 for the investigated complexes is almost the same (Figure 8). The main contribution to LUMO-LUMO + 2 comes from antibonding π*-phenanthroline orbitals and π*-orbitals located along the NO-Ru-X coordinate with antibonding character with respect to the Ru-NO bond. LUMO + 3 in both complexes is formed by an antibonding π*-orbital along the Cl-Ru-N coordinate.

Due to the larger ionization potential of a fluorine atom compared with an oxygen atom, the HOMO orbitals, including contributions of atoms from the NO-Ru-X coordinate, have lower relative energies in the [RuNO(Py)(Bpy)ClF]^+^ complex. This is reflected in the partial inversion of the orbital energies. In pairs 104–103 and 101–100, the orbitals with more contributions from organic ligands (less contribution from F-Ru-NO) are higher in energy in the fluorine complex. Moreover, the total contribution of X-Ru-NO atomic orbitals in HOMO-HOMO-4 is also lower for the fluorine complex compared with the hydroxyl complex. Thus, in the case of the fluorine complex [RuNO(Py)(Bpy)ClF]PF_6_, the transitions in the range of 350–500 nm have more pronounced ligand-to-ligand character, and the charge transfer to NO mainly occurs from organic ligands (bipyridine). For the [RuNO(Py)(Bpy)ClOH]PF_6_ complex, the higher contribution of metal-to-ligand (Ru to NO) charge transfer additionally weakens the Ru-NO bond.

## 3. Materials and Methods

All reagents and sovents were of standard pure grade. The starting ruthenium compound trans(NO,OH),trans(Cl,Cl)-[RuNO(Py)_2_Cl_2_OH] was prepared according to previously described methodology [39]. Elemental analysis (C, H, and N) was performed on a CE-440 analyzer (Exeter Analytical, Coventry, UK).

### 3.1. Synthesis of [RuNO(Py)(Bpy)ClOH]PF_6_ (***1***)

A sample of [RuNO(Py)_2_Cl_2_OH] (0.38 g) and 2,2′-bipyridine (0.23 g) was stirred in water–ethanol mixture (20 mL of water and 50 mL of ethanol) under heating to 80–90 °C for 1.5 h. The complete dissolution of the initial complex was observed after 20–30 min. After heating, the solution was evaporated to ~10 mL, cooled to room temperature, and excess NaPF_6_ was added. The viscous red-orange precipitate was filtered, dried in the air flow, and washed with methylene chloride until the bright red color of the washing solution disappeared. The final orange precipitate was dried in the air flow. The final yield was 66%. Analytical data for C_15_H_14_Cl_1_N_4_O_2_P_1_F_6_Ru_1_ (Mol. Wt. 563.8), calc/found: N—9.9/9.4 C—31.9/31.6 H—2.5/2.2. FT-IR (KBr disk, cm^−1^): 3553 (br) ν(OH); 1852 (s) ν(N=O); 762 (s), 688 (s) ν(PF_6_). Single crystals suitable for SCXRD were prepared via slow diffusion of diethyl ether in the complex solution of acetonitrile.

### 3.2. Synthesis of [RuNO(Py)(Bpy)ClF]PF_6_ (***2***)

A 45 mg sample of [RuNO(Py)(Bpy)ClOH]PF_6_ was dissolved in a minimal amount of HF (72% aqueous solution ~5 mL) and heated in a closed polytetrafluoroethylene (PTFE) beaker at 80–90 °C for 24 h. Afterward, the solution was evaporated to dryness, and the solid residue was dissolved in water and precipitated with NaPF_6_. The precipitate was washed with diethyl ether and dried in the air flow. The final yield was 62%. Analytical data for C_15_H_14_Cl_1_N_4_O_1_P_1_F_7_Ru_1_ (Mol. Wt. 565.8), calc/found:%: N—9.9/9.3 C—31.8/31.7 H—2.3/2.2. FT-IR (KBr disk, cm^−1^): 1917 (s) ν(N=O); 762 (s), 688 (s) ν(PF_6_). Single crystals suitable for SCXRD were prepared via slow diffusion of diethyl ether in the complex solution in acetonitrile.

### 3.3. IR-Spectroscopy

IR-spectroscopy measurements with irradiation were performed using a Bruker VERTEX 80v (Bruker, Billerica, MA, USA) spectrometer with a resolution of 2 cm^−1^ in the range of 400–4000 cm^−1^. The sample (approximately 1–2 mg) was mixed with KBr (around 200 mg) and pressed into common pellets for IR measurements. The pellets were bonded onto the cold finger of a closed-cycle cryostat (Oxford Optistat, Oxford, UK) to control the temperature in the range of 80–300 K. The irradiation procedures were performed using light-emitting diodes (405, 450, 488 nm) through KBr windows perpendicular to the samples with 50–200 mW of optical power. The light power was controlled by Thorlabs optical power meter PM16-401 (Thorlabs, Newton, MA, USA). To investigate the isothermal kinetics of MS1 decay, the following protocol was followed: in the first stage, the sample was irradiated at 80 K with a 450 nm LED (Thorlabs, Newton, MA, USA) in order to generate a sufficient amount of MS1 isomer; subsequently, the temperature was increased to the desired value, and repeated measurements was performed in isothermal mode.

### 3.4. Differential Scanning Calorimetry (DSC)

NETZSCH DSC 204 F1 Phoenix (NETZCH, Selb, Germany) was used to study the kinetics and thermal effects accompanying the phenomenon of the reversible photo-induced transition. DSC measurements of 3–5 mg powdered samples were performed in open aluminum crucibles using the heat flow measurement method at different heating rates of 6, 9, and 12 K/min in 25 mL/min Ar flow. For MS1 generation, the sample was irradiated at 80 K on liquid nitrogen with 445 nm LED for 20 min and quickly transferred to the device chamber. In order to increase the accuracy, measurements were carried out without supply of gas/liquid nitrogen in the measurement cell during the experiment. The sensitivity calibration of the sample carrier sensors and temperature scale graduation were performed by melting and crystal-to-crystal transition measurements of standard samples (C_6_H_12_, Hg, KNO_3_, and In). Processing of the experimental data was performed with Netzsch Proteus Analysis 7.1 software.

### 3.5. Single-Crystal X-ray Diffraction 

SCXRD data were collected using a Bruker Apex Duo diffractometer (Bruker, Billerica, MA, USA) with CCDs using graphite-monochromated MoKα radiation (λ = 0.71073 Å) via 0.5° ω- and φ-scan techniques. Experimental data reduction was performed using the APEX2 suite. The structures were solved using SHELXT and refined by the full-matrix least-squares technique SHELXL [40]. Atomic displacement parameters of the non-H atoms were refined in an anisotropic approximation. Hydrogen atoms of the organic ligands were located geometrically and refined using the riding model. Crystal data for C_15_H_14_ClF_6_N_4_O_2_PRu(1) (M =563.79 g/mol): monoclinic, space group P2_1_/n, a = 12.1474(4) Å, b = 10.9884(4) Å, c = 14.9963(4) Å, β = 103.3930(10)°, V = 1947.27(11) Å^3^, Z = 4, T = 173.15 K, μ(MoK_α_) = 1.101 mm^−1^, D_calc_ = 1.923 g/cm^3^, 14,753 reflections measured (4.64° ≤ 2Θ ≤ 55.018°), 4461 unique (R_int_ = 0.0425, R_sigma_ = 0.0391), which were used in all calculations. The final R_1_ was 0.0301 and wR_2_ was 0.1380 (I > 2σ(I)). Crystal data for C_15_H_13_N_4_ORuClF_7_P (2) (M = 565.78 g/mol): orthorhombic, space group P2_1_2_1_2_1_, a = 7.7613(2) Å, b = 8.8850(2) Å, c = 28.5042(6) Å, V = 1965.63(8) Å^3^, Z = 4, T = 173.15 K, μ(MoK_α_) = 1.095 mm^−1^, D_calc_ = 1.912 g/cm^3^, 14,274 reflections measured (4.802° ≤ 2Θ ≤ 52.764°), 4014 unique (R_int_ = 0.0478, R_sigma_ = 0.0471), which were used in all calculations. The final R_1_ was 0.0316 and wR_2_ was 0.0589 (I > 2σ(I)). CCDC 2348590-2348591 contains the supplementary crystallographic data. These data can be obtained free of charge via http://www.ccdc.cam.ac.uk, accessed on 1 August 2024, or e-mail.

### 3.6. Density Functional Theory (DFT) Calculations

DFT calculations of ruthenium complexes were performed in the ADF2022.105 package [41]. For geometry optimization and calculation of IR frequencies of the investigated complexes, the GGA functional (including the local exchange expression correction by Becke and the local correlation correction by Perdew) [42,43] with QZ4P basis set [44] was used. This basis-functional combination was selected based on previous investigations of similar complexes [45]. For calculation of UV-vis spectra, the structures were previously optimized in GGA functional with QZ4P basis set using COSMO solvation method with water as solvent. Computation of UV-vis spectra was performed at the same level of theory using Davidson method for the first 100 excitations with COSMO solvation applied.

### 3.7. The Hirshfeld Surfaces

Hirshfeld analysis was performed using Crystal Explorer [46]. This program allows the normalized contact distance d_norm_ to be mapped onto the generated Hirshfeld surface. It is customary to map d_norm_ using a red–white–blue scheme, where red denotes close intermolecular contacts (negative d_norm_), blue denotes longer contacts (positive d_norm_), and white denotes intermolecular contacts equal to the van der Waals radii of atoms in contact (d_norm_ = 0). It is possible to obtain two-dimensional plots (fingerprint plots) from the surfaces mapped with d_norm_ values. Derived from the Hirshfeld surface, these 2D fingerprint plots provide a visual summary of the frequency of each combination of d_e_ (radius of external atom) and d_i_ (radius of internal atom) across the surface of a molecule, so they not only indicate which intermolecular interactions are present, but also the relative area of the surface corresponding to each kind of interaction. Points on the plot with no contribution on the surface are left uncolored, points with a contribution to the surface are colored blue for a small contribution, and points with the largest contribution are colored green to red.

## 4. Conclusions

The reaction of trans(NO,OH)-trans(Cl,Cl)-[RuNO(Py)_2_Cl_2_OH] with bipyridine can be used to prepare the heteroleptic pyridine-bipyridine ruthenium nitrosyls [RuNO(Py)(Bpy)ClX]^+^ with a trans NO-Ru-X coordinate. Both obtained complexes (X = OH, F) have a chiral coordination sphere, and the final products [RuNO(Py)(Bpy)ClX]PF_6_ are racemates in their crystal structure. MS1 linkage isomers with the oxygen-coordinated NO group can be prepared for both complexes after irradiation with a 405–450 nm wavelength. The formation of linkage isomers was confirmed by the appearance of new ν(NO) bands after irradiation shifted for 130–150 cm^−1^ from the ground state (GS) band corresponding to the nitrogen-coordinated NO. The achieved MS1 population under equal conditions estimated from the decrease in GS ν(NO) was 17% and 1% for X = OH and F, respectively. The difference can be determined by the nature of HOMO-HOMO-4 orbitals involved in the transitions in the chosen wavelength range. For the hydroxyl complex [RuNO(Py)(Bpy)ClOH]PF_6_, the contribution of MLCT (Ru to NO) is more pronounced. The thermal stability of the MS1 isomer for [RuNO(Py)(Bpy)ClOH]PF_6_ was investigated using DSC and IR techniques, with the characteristic decay temperature (T_d_) being equal to 198–202 K. For further investigations of ruthenium nitrosyls with high decay temperatures of MS1, focusing on electron-accepting ligands with N,O,F-donor atoms seems to hold the greatest prospects.

## Figures and Tables

**Figure 1 molecules-29-04039-f001:**
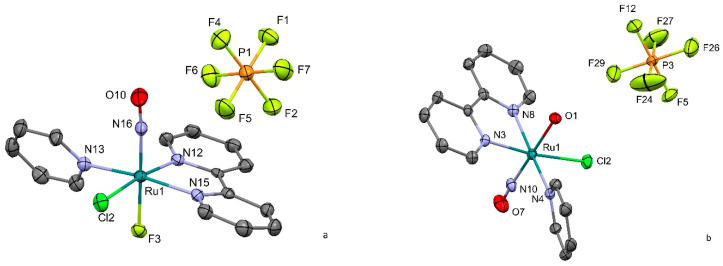
Cation and anion subunits in the crystal structure of [RuNO(Py)(Bpy)ClF]PF_6_ (**a**) and [RuNO(Py)(Bpy)ClOH]PF_6_ (**b**). Thermal ellipsoids are used for 50% probability.

**Figure 2 molecules-29-04039-f002:**
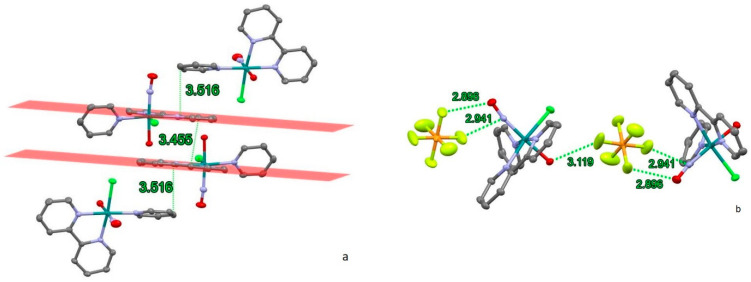
[RuNO(Py)(Bpy)ClOH]PF_6_: (**a**) The stacking interactions; (**b**) closest contacts of NO group. Hydrogen atoms (**a**,**b**) and hexafluorophosphate anions (**a**) are omitted for clarity.

**Figure 3 molecules-29-04039-f003:**
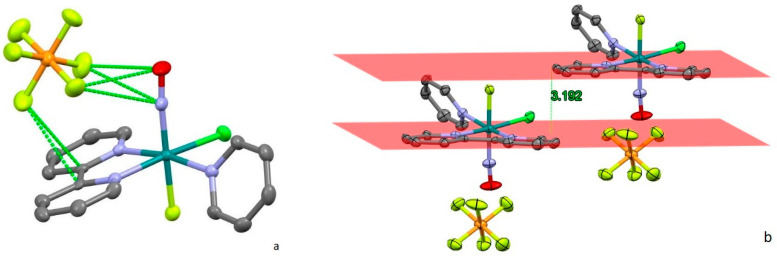
(**a**) Short contacts of anion in unit cell of [RuNO(Py)(Bpy)ClF]PF_6_. (**b**) Spatial arrangement of two adjacent cations. Hydrogen atoms are omitted for clarity.

**Figure 4 molecules-29-04039-f004:**
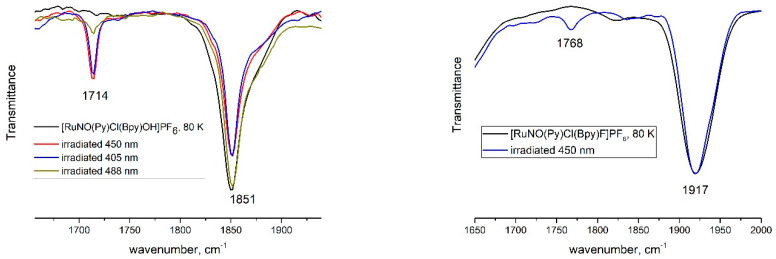
Solid-state IR spectra of complexes **1** and **2** measured at 80 K before and after light irradiation.

**Figure 5 molecules-29-04039-f005:**
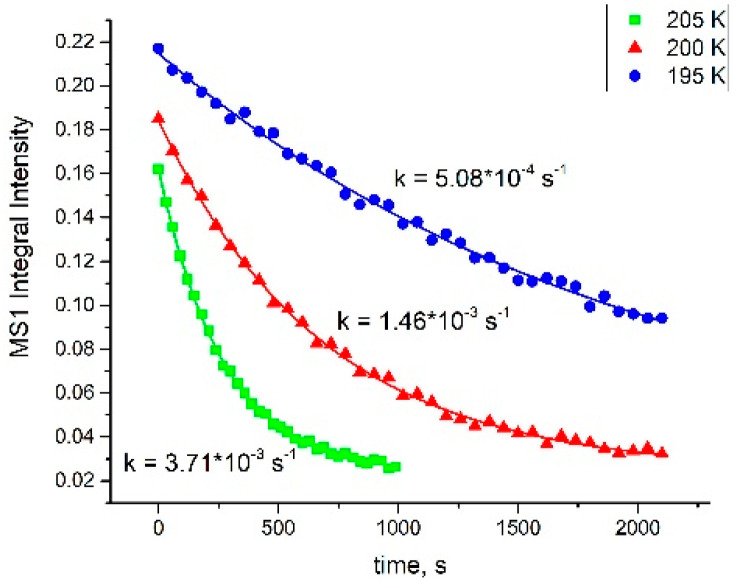
The MS1 decay isothermal kinetic for 1714 cm^−1^ band of [RuNO(Py)(Bpy)ClOH]PF_6_ detected by IR spectroscopy.

**Figure 6 molecules-29-04039-f006:**
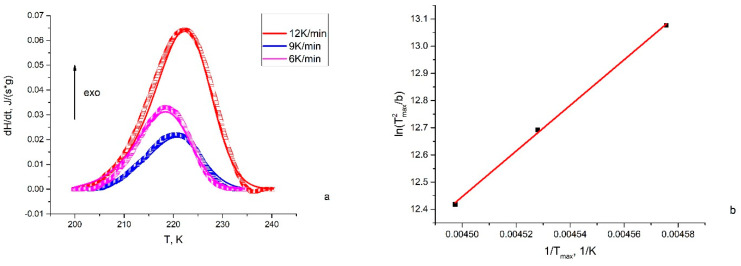
(**a**) Experimental (squares) and calculated (lines) DSC curves for MS1-GS transformation of [RuNO(Py)(Bpy)ClOH]PF_6_; (**b**) Kissenger plot of T_max_-b dependence.

**Figure 7 molecules-29-04039-f007:**
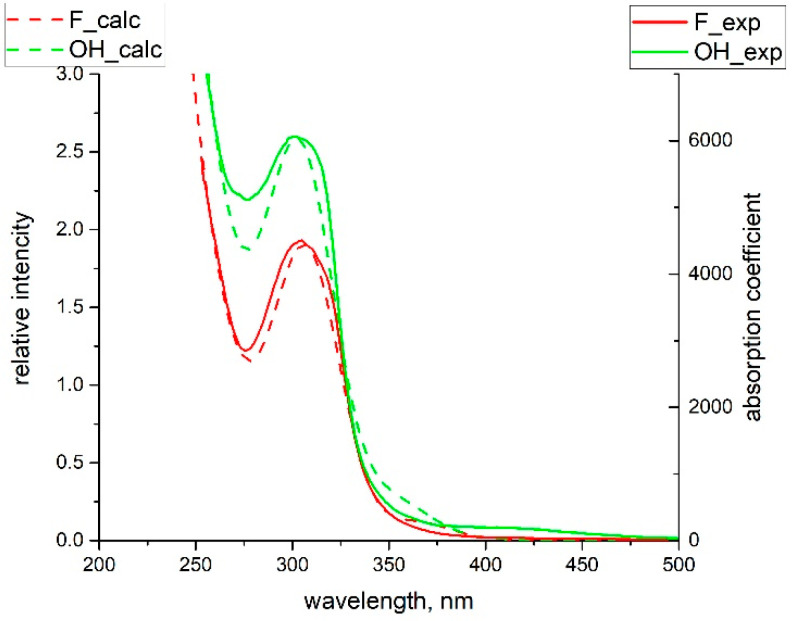
Calculated (lines) and experimental (dash) spectra of water solutions of investigated complexes: [RuNO(Py)(Bpy)ClOH]PF_6_ and [RuNO(Py)(Bpy)ClF]PF_6_.

**Figure 8 molecules-29-04039-f008:**
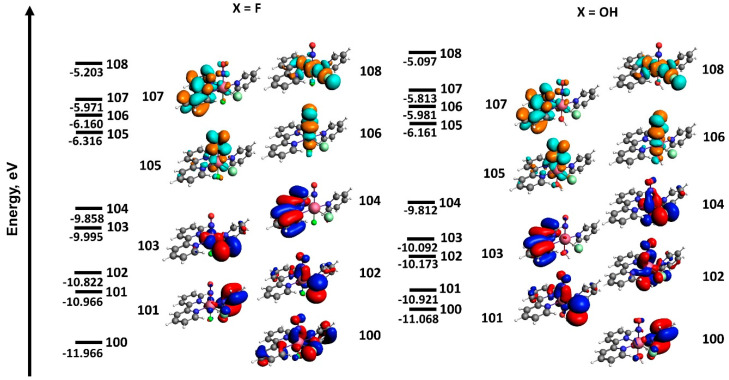
The main orbitals involved in the 350–500 nm transitions in [RuNO(Py)(Bpy)ClX]^+^. The energy of the orbitals is shown schematically.

**Table 1 molecules-29-04039-t001:** The selected geometry parameters in complexes **1** and **2**.

Parameter	[RuNO(Py)(Bpy)ClOH]PF_6_ (1)	[RuNO(Py)(Bpy)ClF]PF_6_ (2)
d(Ru–Cl), Å	2.361(1)	2.371(1)
d(Ru–N_Py_), Å	2.108(2)	2.103(4)
d(Ru–N_bpy_), Å	2.078(2)–2.082(2)	2.063(4)–2.070(4)
d(Ru–NO), Å	1.758(2)	1.727(4)
d(Ru–X), Å (X = F, OH)	1.936(2)	1.948(3)
d(N–O), Å	1.153(3)	1.154(5)
∠Ru–N–O, °	170.6(2)	174.4(4)
∠X-Ru-Cl	88.51(2)	88.97(3)
∠X-Ru–N_Py_	88.12(2)	87.03(2)
∠X-Ru–N_bpy_	85.17–85.54	84.92–85.44
∠Bpy-Py, °	56.07	53.26

**Table 2 molecules-29-04039-t002:** TDDFT-calculated transitions (340–450 nm) in electronic spectra of **1** and **2**.

[RuNO(Py)(Bpy)ClF]PF_6_			[RuNO(Py)(Bpy)ClOH]PF_6_		
Wavelength, nm	Transition occ.-virt.	Contribution, %	Wavelength, nm	Transition occ.-virt.	Contribution, %
428.4	103–105	92	430.4	104–105 102–105 102–106	51 19 7
422.7	103–106	94	424.3	102–105 104–105	52 33
395.5	102–105 102–106 104–105	39 18 11	417.2	104–106 102–106	71 21
367.9	104–105 103–108 102–106	57 20 10	373.2	101–105 102–106 103–105	34 22 11
363.0	103–108 104–105	61 19	370.6	104–108 101–106 102–108	40 14 12
358.6	104–106 104–105	80 7	362.7	101–106 104–108 103–106	37 21 7
352.6	102–106 102–105 100–106	41 24 10	350.8	103–105	80
344.1	100–105 100–106 102–105	20 19 9	340.0	103–106 101–106	84 10

## Data Availability

Data will be made available upon reasonable request.

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
