# Peer review of "Heteroleptic Complexes of Ruthenium Nitrosyl with Pyridine and Bypiridine—Synthesis and Photoisomerization"

_molecules, 2024, doi:10.3390/molecules29174039_

Round 1

Reviewer 1 Report

Comments and Suggestions for Authors

In this manuscript, the authors synthesized two Ru-nitroso complexes and studied their isomerization under light irradiation using infrared spectroscopy. Meanwhile, theoretical calculations revealed the isomerization differences between two complexes. However, there are still some unresolved issues. Therefore, recommend it published after major revision.

1.      How does isomerization caused by light irradiation affect the luminescence properties of complexes?

2.      What is the isomerization process caused by light irradiation, and can the irradiated single crystal structure be obtained?

3.      The authors synthesized complexes coordinated with OH and F respectively. Can nitrogen-containing ligands be attempted to explore the effect of ligand polarity on photoisomerization?

Author Response

Comment 1: How does isomerization caused by light irradiation affect the luminescence properties of complexes?

Response 1:  In general NO group is the typical luminescence quencher and ruthenium nitrosyls with direct coordination of NO to Ru do not show luminescence properties. Probably the irradiation changing NO coordination type can somehow influence the luminescence (f.e. luminescence can occurs in metastable isomers) but up to now no evidences for this were found. 

Comment 2. What is the isomerization process caused by light irradiation, and can the irradiated single crystal structure be obtained?

Response 2: Exactly. If one can achieve the relatively large population of MS by irradiation of single crystal the crystal structure of MS can be refined directly from SCXRD. There are two main problems: 1. Relatively large population can be achieved or not, depending on the photochemical equilibrium. 2. In case of MS1 it is difficult to distinguish MS1 and GS states since N and O are differ only for 1 electron. Still such investigations are possible though we did not provide it because of the low  populations of MS.

Comment 3. The authors synthesized complexes coordinated with OH and F respectively. Can nitrogen-containing ligands be attempted to explore the effect of ligand polarity on photoisomerization?

Response 3. Certainly. Nitrogen ligands such as NH3,  NH2R (R – not arene cycle) can be introduced in trans-position to NO to explore the polarity effects. And in general it is similar to OH, H2O. Some examples were investigated by Kushch and Emelyanov in the beginning of century. We modified conclusion to mention the possibility on N – donor ligands.

Reviewer 2 Report

Comments and Suggestions for Authors

The authors present their work on the synthesis and photochemical behavior of ruthenium nitrosyl complexes.  The work appears to be carefully carried out and the results are sound.  I have just a few suggestions for improvement prior to acceptance:

1. "Complexes" is spelled incorrectly in the title

2.  Line 88: a more complete description of the HF used should be included (i.e., % HF solution? aqueous solution?)

3.  Line 139: the reason for selection of the specific DFT functional and basis set should be included (i.e., has this combination been used successfully before for similar complexes?)

4.  Figure 1: The % thermal ellipsoid probability should be included in the caption.

5.  Figure 1: Please label ALL of the non-carbon atoms for easier identification.

6.  Figure 5: it should be specified in the caption that it is the NO bond at 1714 cm-1 that is being followed.

7.  While the DFT calculations provide the reason for MS1 production via HOMO/LUMO interactions, the authors do not comment on the absence of MS2 production (as discussed in the intro) if it is relevant to these complexes.

Great work and I look forward to seeing the final publication.

Comments on the Quality of English Language

none

Author Response

  1. "Complexes" is spelled incorrectly in the title

Response 1. Corrected.

  1.  Line 88: a more complete description of the HF used should be included (i.e., % HF solution? aqueous solution?)

Response 2. 72 % aqueous solution of HF was used. Added to the text. 

  1. Line 139: the reason for selection of the specific DFT functional and basis set should be included (i.e., has this combination been used successfully before for similar complexes?)

Response 3. Indeed the combination was used before and seemed to be the most reasonable. Added to text and one additional reference  (45) added.

  1. Figure 1: The % thermal ellipsoid probability should be included in the caption.

Response 4. Done according to the referee comment.

  1. Figure 1: Please label ALL of the non-carbon atoms for easier identification.

Response 5. Done according to the referee comment.

  1. Figure 5: it should be specified in the caption that it is the NO bond at 1714 cm-1 that is being followed.

Response 6. Done according to the referee comment.

  1. While the DFT calculations provide the reason for MS1 production via HOMO/LUMO interactions, the authors do not comment on the absence of MS2 production (as discussed in the intro) if it is relevant to these complexes.

Response 7.

«The generation of MS2 isomer is usually performed by IR-irradiation of MS1 since the UV spectra of GS and MS2 are strongly overlapped and MS2-MS1 photochemical equilibrium is shifted to MS1 isomer in the blue region. Still some examples of MS2 formation together with MS1 after blue-light irradiation of GS were detected. Due to the structure features the shift of n(NO) band in GS-MS2 transition (270-350 cm-1) is bigger than for GS-MS1 transition (130-150 cm-1) and in investigated complexes no corresponding changes in IR-spectra were found.» - The text was added. And the reference (30) was also added.  

So we did not detected MS2 in experiments that is why we do not try to calculate the corresponding state.